# Thermodynamics, Kinetics, and Mechanisms of the Co-Removal of Arsenate and Arsenite by Sepiolite-Supported Nanoscale Zero-Valent Iron in Aqueous Solution

**DOI:** 10.3390/ijerph191811401

**Published:** 2022-09-10

**Authors:** Meihaguli Ainiwaer, Xibai Zeng, Xianqiang Yin, Jiong Wen, Shiming Su, Yanan Wang, Yang Zhang, Tuo Zhang, Nan Zhang

**Affiliations:** 1Key Laboratory of Agro-Environment, Institute of Environment and Sustainable Development in Agriculture, Chinese Academy of Agriculture Sciences, Ministry of Agriculture and Rural Affairs, Beijing 100081, China; 2College of Natural Resources and Environment, Northwest A&F University, Yangling, Xianyang 712100, China; 3Scientific Observation and Experiment Station of Yueyang, Ministry of Agriculture, Yueyang Agricultural Research Academy, Yueyang 414021, China; 4College of Environmental Science & Engineering, China West Normal University, Nanchong 637009, China

**Keywords:** adsorption, sepiolite, nZVI, arsenate, arsenite

## Abstract

In this study, a newly synthesized sepiolite-supported nanoscale zero-valent iron (S-nZVI) adsorbent was tested for the efficient removal of As(III) and As(V) in aqueous solution. Compared with ZVI nanoparticles, the As(III) and As(V) adsorption abilities of S-nZVI were substantially enhanced to 165.86 mg/g and 95.76 mg/g, respectively, owing to the good dispersion of nZVI on sepiolite. The results showed that the adsorption kinetics were well fitted with the pseudo-second-order model, and the adsorption isotherms were fitted with the Freundlich model, denoting a multilayer chemical adsorption process. The increase in the initial solution pH of the solution inhibited As(III) and As(V) adsorption, but a weaker influence on As(III) than As(V) adsorption was observed with increasing pH. Additionally, the presence of SO_4_^2−^ and NO_3_^−^ ions had no pronounced effect on As(III) and As(V) removal, while PO_4_^3−^ and humic acid (HA) significantly restrained the As(III) and As(V) adsorption ability, and Mg^2+^/Ca^2+^ promoted the As(V) adsorption efficiency. Spectral analysis showed that As(III) and As(V) formed inner-sphere complexes on S-nZVI. As(III) oxidation and As(V) reduction occurred with the adsorption process on S-nZVI. Overall, the study demonstrated a potential adsorbent, S-nZVI, for the efficient removal of As(III) and As(V) from contaminated water.

## 1. Introduction

Arsenic (As) is a toxic metalloid that is widely distributed in the environment and is known to be a serious threat to food safety and human health [1,2,3,4]. Numerous health effects are associated with As exposure and As can cause severe health problems, such as liver, lung, kidney, and skin cancers [5,6,7,8]. Arsenic exposure pathways from the environment normally include consumption of contaminated food and water, inhalation of dust, and incidental ingestion of soil. In some well-documented regions (e.g., Bangladesh; West Bengal, India; and Shimen County, China), consumption of As-contaminated drinking water and crops represents the major exposure sources and pathways [9,10,11]. Hence, arsenic has been categorized as the first priority issue among toxic substances, and a maximum contamination level of 10 μg·L^−1^ for As in drinking water was established by the World Health Organization (WHO) [8]. Arsenic is most commonly found in two inorganic forms in the environment: arsenate (As(V)) and arsenite (As(III)). This normally depends on the environmental conditions, e.g., pH and redox potential (Eh). In groundwater with a pH range of 6–8.5, As(III) dominates as the H_3_AsO_3_ species under reducing conditions (low Eh values), while As(V) appears as a mixture of the H_2_AsO_4_^−^ and HAsO_4_^2−^ species under oxidizing conditions (high Eh values) [12]. However, their toxicity and adsorption affinity are significantly different: As(III) is more mobile, soluble, and toxic than As(V) and is more difficult to remove by adsorbents than As(V) [13,14]. It is noteworthy that nanoparticle size adsorbent displayed a ~10% greater removal efficiency (RE) for As(III) than for As(V) under certain experimental conditions [15].

Arsenic removal by adsorbents largely depends on solution pH, increasing the difficulty of wastewater disposal considering the wide pH range of contaminated water. However, it is relatively difficult to remove arsenite (As(III)), which can remain in the form of H_3_AsO_3_ when the pH is lower than 9.2 [16,17]. Previous research indicated that uncharged H_3_AsO_3_ is more difficult to remove using traditional physical-chemical treatment methods [18]. Therefore, it is necessary to seek integrated materials that exhibit highly efficient As(III) oxidation and adsorption for the removal of As(III) from arsenic-polluted water, and additional oxidation is often required for conversion from As(III) to As(V).

There are many methods for removing heavy metals, such as coagulation, ion exchange, adsorption, membrane filtration, and electrochemistry; among them, adsorption is widely used because of its simple operation and economic and feasible characteristics [19,20,21]. Many studies have focused on iron oxides and their precursors, such as nano zero-valent iron (nZVI). Gil-Diaz et al. [22] found that when nZVI was employed for in situ remediation of arsenate in soil, both oxidation and adsorption of As(III) happened at the same time. The attractive properties of nanoadsorbents used for arsenic removal from water include a high surface area and high reactivity [23]. However, as an adsorbent for arsenic removal in wastewater, nZVI suffers from the drawback of easy aggregation. Therefore, supporting substances such as zeolite, biochar, and sepiolite are usually employed to improve the dispersion of nZVI nanoparticles [24,25,26]. Sepiolite (SEP) is an abundant natural clay mineral with a stable structure and low cost, and it also exhibits remarkable effectiveness in removing As from waste water [27]. Its polyporous structure is a good choice for dispersing the nZVI homogeneously in the aqueous solution, which can maximize the effectiveness of nZVI. Contaminants including lead, chromium, and metoprolol were removed using sepiolite-supported nZVI (S-nZVI) from aqueous solution in previous studies [28,29]. However, less attention has been devoted to the removal ability of As(III) and As(V) and the underlying mechanism of arsenic adsorption by S-nZVI.

The objective of the present study was to prepare different S-nZVI adsorbents with the highest removal capacity for As(III) and As(V). The factors influencing the arsenic adsorption efficiency were well explored and included pH, dosage, and coexisting ions. In addition, the adsorption kinetics and isotherms of As(III) and As(V) were investigated. Finally, the mechanisms of As(III)/As(V) removal were interpreted with the help of X-ray photoelectron spectroscopy (XPS) and X-ray diffraction (XRD) analysis. This study will contribute to the knowledge on the application of S-nZVI for the remediation of As(III) and As(V) in contaminated water.

## 2. Materials and Methods

### 2.1. Chemicals

All chemical reagents used in this study were of analytical grade. The sepiolite was obtained from Neixiang, Henan Province, China. The sepiolite was ground and passed through a 0.15 mm pore size sieve. Stock solutions of As(III) and As(V) were prepared by dissolving appropriate amounts of NaAsO_3_ and Na_2_HAsO_4_.12H_2_O in deionized water (Millipore Milli-Q 18 MX).

### 2.2. Synthesis of Materials

Acid-activated sepiolite (ASEP) was prepared according to Zhang et al. [30]: natural sepiolite (SEP) was immersed in HCl solution (2 mol/L) at 80 °C for 8 h under magnetic stirring [30]. The synthesis process of the S-nZVI composite was simplified and improved based on the method mentioned in a previous study [24]. Briefly, 8 g of ASEP and 200 mL of FeCl_3_·6H_2_O solution were mixed in a 1000 mL three-necked flask. The mass ratios of Fe:sepiolite were set to 1:3, 1:9, and 1:15. All operations were conducted under room temperature and pressure conditions. After 15 min of ultrasound treatment, the mixture was stirred for another three hours. Subsequently, 200 mL of 0.98 mol/L NaBH_4_ was added dropwise with stirring to perform the reduction reaction, and a black mixture of S-nZVI was formed. The reduction reaction is as follows (1):4Fe^3+^ + 3BH_4_^−^ + 9H_2_O → 4Fe^0^ + 3H_2_BO_3_^−^ + 6H_2_ +12H^+^(1)

Then, the S-nZVI was separated from the mixture by a magnetic field, washed with deionized water and absolute ethanol, freeze-dried, and stored under dry conditions. The nZVI particles were prepared as described above, except that absolute ethanol was added during stirring to control the sizes of the nanoparticles.

### 2.3. Characterization of Samples

The solid phases were characterized through XRD using a Rigaku Ultima IV (Japan) device with Cu Kα radiation (λ = 0.1541 nm) at a voltage and current of 40 kV and 40 mA, respectively. A scan range between 2° and 70° and a step size of 0.02° were used, with a change rate of 5 s/step. The infrared spectra of each sample were collected using a Nicolet 6700 FTIR spectrometer (Thermo Fisher, Waltham, MA, USA) within a range of 4000–400 cm^−1^. The specific surface area (SSA) and pore volume were estimated by Brunner–Emmet–Teller (BET) analysis (Belsorp-Mini II, Microtracbel Inc., Osaka, Japan). The valence states of the composites were determined by performing XPS with monochromatic Al Kα radiation (1486.8 eV) by using an Amicus XPS instrument (Shimazu Corp., Kyoto, Japan). The morphology and elemental composition of the selected samples were examined using scanning electron microscopy (SEM) (SU8100, HITACHI Corp., Tokyo, Japan).

### 2.4. Batch Sorption Experiments

#### 2.4.1. Optimization of the Adsorbent Preparation Conditions

To obtain S-nZVI with the highest RE for arsenic, S-nZVI was prepared with different mass ratios of Fe:sepiolite as the starting material. Then, the arsenic RE of S-nZVI with different Fe:sepiolite ratios was evaluated by adding 30 mg adsorbent to 30 mL arsenic solutions with 0.01 mol/L NaCl (background electrolyte) in a 50 mL centrifuge tube. The mixture was shaken at 180 rpm at 25 °C for 24 h. The initial pH of the solution was adjusted to 7.0 using diluted HCl and NaOH (0.1 mol/L). The adsorption capacity and RE at equilibrium were calculated according to Equations (2) and (3):(2)$$q_{e}=\frac{(C_{0}-C_{e})v}{m}$$
(3)$$RE=\frac{C_{0}-C_{e}}{C_{0}}\times 100\%$$
where *q_e_* is the equilibrium adsorption capacity (mg/g), V is the volume of solution (mL), *C_e_* and *C*_0_ represent the concentrations at equilibrium and the initial time (mg/L), respectively, and m is the weight of the adsorbent (g).

The suspensions were filtered through 0.22 μm filters before being tested. The As concentration was determined by atomic fluorescence spectrometry (AFS-8220, JiTian instrument Co., Ltd, Beijing, China).

#### 2.4.2. Adsorption Kinetics

The adsorption kinetics of As(III) and As (V) by S-nZVI (sepiolite: Fe at a ratio of 3:1) were evaluated by batch adsorption experiments. In general, 1 g/L S-nZVI was dosed into flasks containing 50 mg/L As(III) and As(V) solutions at an initial pH of 7.0. At predetermined time intervals until 24 h, the aliquots of the solution were collected and centrifuged. The mixture was shaken in a thermostatic shaker at 180 rpm and 25 °C. The As(III) and As(V) concentrations in the solutions were tested with the AFS system as mentioned in Section 2.4.1. The pseudo-first-order and pseudo-second-order models can be expressed as Equations (4) and (5):(4)$$q_{t}=q_{e}\left[1-\exp\left(-k_{1}t\right)\right]\;$$
(5)$$\frac{t}{q_{t}}=\frac{1}{k_{2}q_{e}^{2}}+\frac{t}{q_{e}}\;$$
where *q_t_* and *q_e_* (mg/g) are the adsorption quantity at adsorption time (t) and equilibrium, respectively. *k*_1_ and *k*_2_ represent the adsorption rate constants.

#### 2.4.3. Adsorption Isotherm

The adsorption isotherms of S-nZVI were tested by dosing 30 mg adsorbent into flasks containing 30 mL As(III) and As(V) at different concentrations (varying from 5 to 600 mg/L). The mixture of adsorbent and arsenic solution was shaken in a thermostatic shaker at 180 rpm and 25 °C with an initial pH of 7.0 for 14 h. The adsorption isotherms were fitted by the Langmuir (assuming a monolayer adsorption process) and Freundlich models (assuming that multilayer adsorption occurred) [31]. The Langmuir and Freundlich models can be expressed by Equations (6) and (7), respectively:(6)$$q_{e}=\frac{q_{m}k_{L}C_{e}}{1+K_{L}C_{e}}$$
(7)$$q_{e}=K_{f}C_{e}^{\frac{1}{n}}$$
where *C_e_* and *q_e_* are the equilibrium concentration (mg/L) and adsorption amount (mg/g) of heavy metal, respectively; *q_m_* (mg/g) is the maximum adsorption capacity; *K_L_* (L/mg) is the Langmuir constant; and *K_f_* and 1/n are the Freundlich constants.

To evaluate the feasibility of the adsorption process, R_L_ was calculated according to Equation (8) [32]:(8)$$R_{L}=\frac{1}{1+K_{L}*C_{0}}$$

The adsorption isotherms were also fitted with the Dubinin–Astakhov (D-A) model and Dubinin–Radushkevich (D-R) model. The Dubinin–Astakhov (D-A) model was expressed as Equations (9) and (10) [33,34]
(9)$$\varepsilon =\mathrm{RT}\ln\frac{C_{s}}{C_{e}}$$
(10)$$q_{e}=q_{mD-A}e^{\left[-{\left(\frac{\varepsilon }{E_{DA}}\right)}^{n_{DA}}\right]}$$
(11)$$q_{e}=q_{mD-R}e^{-K_{DR}\varepsilon^{2}}$$
where *q_mD-A_* (mg/g) and *q_mD-R_* (mg/g) are the maximum adsorption amount, ε (kJ/mol) is the adsorption potential, Cs (mg/L) is the solubility of arsenate and arsenite, which was 2.46 × 10^5^ mg/L and 9.01 × 10^5^ mg/L for As(V) and As(III) at 25 °C, R (kJ/K/mol) is the gas constant equal to 8.314 × 10^−^^3^, *C_e_* (mg/L) is the equilibrium concentration, *q_e_* (mg/g) is the adsorption amount, T = 273.15 K at the experimental condition, *E_DA_* (kJ/mol) is the characteristic energy, *n_DA_* is a constant related to the percent of pore filling, and KDR (mol^2^/kJ^2^) is the model constant of the D-R model [35].

#### 2.4.4. Effect of Initial pH, Adsorbent Dosage and Coexisting Ions

The influence of the initial pH, adsorbent dosage, and coexisting ions was assessed by examining the equilibrium adsorption amounts of As(III) and As(V) on S-nZVI. For the study of pH influence, the initial pH of the sorption solution was adjusted from 3.0 to 9.0 by adding HCl (0.1 mol/L) or NaOH (0.1 mol/L). The S-nZVI adsorbent was dosed into an arsenic solution of 50 mg/L at a dosage ratio of 1 g/L. For the study of the adsorbent dosage effect, the adsorbent was dosed into arsenic solution at concentrations ranging from 0.2 to 8 g/L. To study the influence of coexisting ions, the S-nZVI adsorbent (1 g/L) was dosed into a mixed solution with 50 mg/L arsenic and each coexisting ion (0.01 mol/L Ca^2+^, Mg^2+^, NO_3_^−^, SO_4_^2−^ or PO_4_^3−^) or 200 mg/L humic acid (HA).

## 3. Results

### 3.1. Adsorption Kinetics of As(III)/As(V) by S-nZVI

The composites were prepared under a graded series of theoretical mass ratios of sepiolite to Fe (3:1, 9:1, and 15:1), and RE was examined based on the adsorption kinetics at pH = 7.0 (Figure 1). Figure 1a shows the As(III) and As(V) adsorption amounts of S-nZVI when the Fe:sepiolite ratio varied from 1:3 to 1:9 and 1:15. When the mass ratio of sepiolite:Fe increased from 3:1 to 15:1, the RE of As(III) decreased from 97.89% to 41.87%, while that of As(V) decreased from 79.33% to 22.00%. The decrease in arsenic removal capacity with the increase in the relative sepiolite content in S-nZVI illustrated that nZVI was the key functional component in removing arsenic. Therefore, in the following experiments, S-nZVI with a sepiolite:Fe ratio of 3:1 was adopted as the adsorbent.

Figure 1b shows the As(III) and As(V) adsorption kinetics by S-nZVI. The adsorption process of both As(III) and As(V) exhibited a tendency of rapid adsorption in the first 2 h and then gradually reached equilibrium by 24 h. The equilibrium adsorption amounts were 38.44 ± 0.59 mg/g and 40.66 ± 0.21 mg/g for As(V) and As(III), respectively, consistent with the result (Figure 1a) that As(III) was more advantageous than As(V) for removal by the S-nZVI adsorbent.

The pseudo-first-order and pseudo-second-order kinetic parameters for adsorption are given in Table 1. For both adsorbates, the equilibrium adsorption quantity calculated from the pseudo-first-order and pseudo-second-order models was very close to that from the experimental data. The adsorption process of both As(III) and As(V) better suited the pseudo-second-order model (R^2^ > 0.99). Better fitting with the pseudo-second-order model indicated that the adsorption of As(III) and As(V) was a chemical adsorption process [36]. In addition, the adsorption rates (k_2_) of As(III) and As(V) were 1.12 g·mg^−^^1^·h^−^^1^ and 0.28 g·mg^−^^1^·h^−^^1^, respectively, indicating that the removal process of As(III) by S-nZVI was more rapid than that of As(V).

### 3.2. Isotherm Adsorption Study of As(III)/As(V) by S-nZVI

The parameters of the As(III) and As(V) adsorption isotherms fitted with the Langmuir and Freundlich models are shown in Appendix A and Table 2. By comparing the R^2^ values of both models, it was found that the Freundlich model can better describe the absorption process of S-nZVI for As(V) (R^2^ = 0.974) while the Langmuir model can better describe the As(III) adsorption by S-nZVI (R^2^ = 0.967). The better fitting results of the Freundlich model suggest that ion sorption on S-nZVI is a multilayer adsorption process. The maximum adsorption capacities of S-nZVI towards As(III) and As(V) was 165.86 mg/g and 95.76 mg/g, respectively, which were much higher than those of other nZVI-based materials reported in previous studies (Appendix A). For example, the As(III) and As(V) adsorption capacity by Mn-nZVI was reported to be 59.90 mg/g and 45.50 mg/g, respectively. This excellent removal ability of As(III) and As(V) indicated that S-nZVI is an effective sorbent for arsenic treatment of contaminated water. Table 2 shows that 1/n was 0.38 and 0.41 for As(III) and As(V), respectively, and RL ranged between 0 and 1, indicating that As(III) and As(V) sorption on S-nZVI is a favorable process [31]. The difference between the adsorption capacities of As(III) and As(V) was most likely associated with the chemical behavior of As(III) and As(V) anions, which might be controlled by diverse adsorption mechanisms [37]. The adsorption isotherms were also fitted with the Dubinin–Astakhov (D-A) model and Dubinin–Radushkevich (D-R) model (Appendix A and Appendix A). The D-A and D-R models can be applied to describe the adsorption process on adsorbents with pores following Gaussian energy distribution [34]. From the fitting results (Appendix A), it can be seen that As(III)/As(V) adsorption isotherms can be interpreted by both models, indicating that adsorption of As(III)/As(V) on S-nZVI happened in homogeneous microporous systems [38].

### 3.3. Influence of Initial pH on As(III)/As(V) Adsorption

The speciation of heavy metal ion species at different pH values can influence the reaction as well. To elucidate the effect of pH on the sorption process, adsorption edge experiments were conducted in the pH range from 3.0 to 9.0 for both As(III) and As(V). Figure 2a,b shows that the As(III) and As(V) sorption capacities of SEP were relatively stable with pH variation but were significantly lower than those of nZVI and S-nZVI, ranging between 5 and 7 mg/g. The adsorption capacities of As(V) and As(III) decreased with increasing initial pH for both nZVI and S-nZVI, which is in accordance with previous studies [39,40]. However, the As(III) adsorption capacity of S-nZVI decreased slowly from 48.71 to 44.74 mg/g when the pH changed from 3 to 9, while the As(V) sorption capacity decreased from 47.72 to 33.33 mg/g. From the chemical species calculation results (Figure 2c,d), it can be seen that As(V) existed in anion form (H_2_AsO_4_^−^ or HAsO_4_^2−^) between pH 3 and 9, while As(III) was mostly an uncharged ion (H_3_AsO_3_) with a small part as a charged anion (H_2_AsO_3_^−^) when the pH was higher than 8. The pH increase led to the deprotonation of the S-nZVI adsorbent, which may lower the electrostatic attraction between the adsorbent and anionic pollutants (H_2_AsO_4_^−^, HAsO_4_^2−^, H_2_AsO_3_^−^). This explained why pH variation had a larger influence on the sorption of As(V) than that of As(III). The dashed lines in Figure 2a,b demonstrate the pH variation before and after As(III) or As(V) adsorption, which reflects the pH buffering capacity of each adsorbent. The deviation of the dashed lines from the line with a slope of 1 and an intercept of 0 (orange dot line) show how much the pH of aqueous solution can be retained after adsorption. Figure 2a,b shows that nZVI and S-nZVI had a higher pH buffering capacity compared with sepiolite adsorbent, possibly due to the nZVI modification.

### 3.4. Influence of Adsorbent Dosage

With the objective of decreasing the arsenic concentration as much as possible in an economical way, it is necessary to optimize the sorbent dosage in practical applications. The effect of the adsorbent dosage was studied for As(III) and As(V) adsorption with the concentration of S-nZVI varying from 0.2 to 8.0 g/L (Appendix A). The results showed that RE increased from 45.14% to 99.76% for As(III) and from 35.39% to 97.51% for As(V) when the S-nZVI dosage increased from 0.2 to 8.0 g/L. When the S-nZVI dose was 1 g/L, the As(III) and As(V) RE were above 80%, while the heavy metal RE almost reached 100% when the dosage was 4 g/L. Comparing As(III) and As(V) adsorption, it can be found that the As(III) RE was higher than that of As(V) at each S-nZVI dosage, reflecting the excellent affinity between the adsorption sites of S-nZVI and As(III). The reason for this phenomenon was that the increased material content led to an increased surface area and number of adsorption sites.

### 3.5. Influence of Coexisting Ions/Substances on As(III)/As(V) Adsorption

Normally, there are various coexisting anions in wastewater, and thus, determining the influence of these ions on arsenic RE is vital for adsorbent development. Figure 3 shows the influence of coexisting ions (0.01 mol/L Ca^2+^, Mg^2+^, NO_3_^−^, SO_4_^2−^, and PO_4_^3−^ and 200 mg/L HA) on the adsorption of As(III) and As(V) by S-nZVI. From Figure 3, it is clear that the ions have different effects on sorption capacity. It can be concluded that the presence of SO_4_^2−^ and NO_3_^−^ had little influence on the adsorption of either As(V) or As(III). The interaction between sulfate/nitrate and nZVI was reported to be due to the formation of outer-sphere complexes, which are not as tightly bound as inner-sphere complexes [41,42]. Therefore, it is difficult for SO_4_^2−^ and NO_3_^−^ to compete for binding sites with arsenic due to their weaker affinity with S-nZVI. On the other hand, the presence of PO_4_^3−^ and HA significantly inhibited the adsorption of As(III) and As(V). The great loss of As(III)/As(V) adsorption capacity by competition with PO_4_^3−^/HA could be due to the similarity of their molecular structures [43,44]. Coexisting HA introduced not only A^−^ but also protons, thus increasing the positive charges on the adsorbent. The decreased arsenic RE due to HA competition illustrated that the increased electrostatic attraction due to introduced protons could not overcome the competition effect of A^−^. Ca^2+^ and Mg^2+^ considerably enhanced the RE of As(V) from 76.88% to 98.36% and 97.73%, while no such promoting effect was found for As(III) adsorption. It is assumed that Ca^2+^/Mg^2+^ can increase the positive charges on the S-nZVI adsorbent, which is beneficial for anion adsorption through the bridging effect. As presented in Figure 2c,d, at the experimental pH value of 7, As(III) was electrically neutral, while As(V) was in anion form. Therefore, the coexistence of Ca^2+^/Mg^2+^ and the change in surface charge showed a greater influence on As(V) adsorption than on As(III) adsorption. This observation was consistent with some previous studies [45,46].

### 3.6. S-nZVI Characterization and Sorption Mechanisms

It was demonstrated that spherical nZVI was well dispersed on the sepiolite surface without agglomeration (Appendix A). The specific surface areas of SEP, ASEP, and S-nZVI were 23.21, 110.00, and 59.01 m^2^/g, respectively. The above results confirmed that nZVI was well loaded on the surface of SEP and that sepiolite could inhibit nZVI agglomeration.

The broad XPS scan indicated that S-nZVI was primarily composed of Fe, O, and Si. Small amounts of C, Ca, and Mg were also detected, which might be contributed by SEP (Figure 4a). The curve fitting of the As3d peaks involved four subpeaks because the As3d peak for As(III) and As(V) has two spin-orbit splittings, denoted as As3d3/2 and As3d5/2, with a constant binding energy gap of 0.7 eV. By comparing the areas of the subpeaks corresponding to As(III) and As(V), the relative abundance of each valence state can be calculated. From Figure 4b, it can be seen that after As(V) was adsorbed on S-nZVI, 53.55% of the total arsenic was detected as As(III), indicating that some As(V) was reduced to As(III). Figure 4c demonstrates that after As(III) was adsorbed on S-nZVI, 39.64% of the total arsenic was oxidized to As(V), while 60.36% of arsenic remained as As(III). The results suggest that As(V) can be reduced by S-nZVI and that As(III) oxidation can happen by reacting with S-nZVI. In other words, the S-nZVI adsorbent served as both a reductant and an oxidant for arsenic [40,47]. The same results were reported in a previous study, in which the arsenic reduction and oxidation mechanism was interpreted to be due to the Fe oxide shell outside the Fe^0^ core of nZVI particles [48]. Electrons from Fe^0^ can be transferred via the oxide layer and lead to the reduction of As(V). On the other hand, the iron oxides on the outer layer account for As(III) oxidation. It was also explained in previous studies that the corrosion of Fe^0^ in solution generated intermediate compounds known as reactive oxygen species (ROS), leading to the oxidation of As(III) [49,50]. This phenomenon is consistent with the Fe2p spectrum. Figure 4d–f represents the valence state change of Fe on S-nZVI before and after As(V) or As(III) adsorption. Before adsorption, iron on the adsorbent surface was largely oxidized to Fe^2+^ and Fe^3+^, with only 10.82% of the total iron remaining as Fe^0^. After As(V) or As(III) adsorption, the ratio of Fe^0^ was further decreased to 3.40% and 4.12%, respectively. Furthermore, the binding energy of Fe^0^, Fe^2+^, and Fe^3+^ was shifted to higher positions after As(III) and As(V) adsorption, suggesting that As(III) and As(V) were adsorbed onto Fe composites and formed inner-sphere complexes [51,52,53]. The Fe^0^ loss of nZVI confirmed the hypothesis that Fe^0^ was oxidized and that an Fe oxide shell was formed on the adsorbent surface during arsenic adsorption. After As(V) was adsorbed on S-nZVI, the ratios of Fe^0^ and Fe^2+^ were determined to be 3.40% and 43.80%, respectively, which are lower than the values of 4.12% and 46.53% for As(III) adsorption. The higher degree of iron oxidation caused by As(V) adsorption indicated that As(V) reduction happened at the same time as adsorption, in agreement with the As3d spectrum.

The spectra of O1s consisted of peaks at ~530 eV, ~531 eV, and ~532.5 eV, which represent the binding energies of O^2−^(M-O), OH-, and physically or chemically adsorbed water (H_2_O), respectively [30]. For the adsorption of As(III) and As(V), the ratio of the peak area for O^2−^ increased from 19.32% to 40.51% and 30.54%, respectively. The variation in O^2−^ may be due to the adsorption of arsenic and the formation of As-O bonds on S-nZVI. The higher peak area ratio of O^2−^ for As(III) also proved that the As(III) adsorption capacity was higher than that of As(V). Furthermore, the percentage of -OH groups on S-nZVI increased from 22.64% to 42.09% and 44.89% after As(III) and As(V) adsorption, respectively. Thus, it is reasonable to conclude that the increase in the intensity of the -OH peak was due to surface adsorption reactions on the surface of S-nZVI [1].

Appendix A shows the XRD patterns of S-nZVI before and after adsorption of As(III) and As(V). After arsenic adsorption, the peak for Fe^0^ at 44.68^°^ was largely eliminated, indicating the oxidation of Fe^0^ on S-nZVI during the adsorption process. The presence of the characteristic Fe_2_O_3_ peak (35.5°) on S-nZVI before and after arsenic adsorption corresponded to the oxidation of Fe^0^. From the patterns after As(V) adsorption, FeAsO_4_ (59.05°) precipitation was shown to be formed on S-nZVI [31]. In addition to FeAsO_4_ precipitation, Fe_2_As_4_O_12_ (21.05°) was observed to be formed on S-nZVI after As(III) adsorption, proving that As(III) oxidation simultaneously happened with adsorption. In accordance with the XPS Fe2p spectrum, the formation of Fe_2_As_4_O_12_ indicates the existence of Fe(II) during As(III) adsorption and oxidation. The spectral analysis provided evidence that the mechanism of As(III) and As(V) removal by S-nZVI involves surface precipitation and inner-sphere complexation. Furthermore, the adsorption of As(III) and As(V) was accompanied by the oxidation/reduction process of iron and arsenic [1,52,54,55].

## 4. Conclusions

In this study, a sepiolite-supported nanoscale zero-valence iron adsorbent (S-nZVI) was successfully prepared and utilized in As(III) and As(V) adsorption from aqueous solution. An adsorption isothermal study was carried out. Removal of both As(III) and As(V) followed pseudo-second-order kinetics with a higher rate constant for As(III) than As(V), revealing a faster adsorption process for As(III). As(III) and As(V) adsorption isotherms were well fitted by the Freundlich isotherm model, and the maximum adsorption capacity of S-nZVI for As(III) and As(V) was 165.86 mg/g and 95.76 mg/g, respectively. In addition, increases in solution pH substantially inhibited the adsorption capacity of As(III) and As(V) by S-nZVI, with a stronger influence on As(V) being observed. Coexisting PO_4_^3−^ and humic acid (HA) competed strongly with the adsorption of As(III) and As(V) on S-nZVI, while the influence of SO_4_^2−^ and NO_3_^−^ was negligible. The presence of Ca^2+^ and Mg^2+^ in solution enhanced only the As(V) adsorption ability, with little effect on As(III) adsorption. XPS As3d, Fe2p, and O1s analysis confirmed that the mechanisms of As(III) and As(V) adsorption include precipitation and surface complexation, which means that Fe-O bonds were formed between As(III)/As(V) and S-nZVI. Furthermore, the oxidation and reduction of arsenic happened simultaneously with the adsorption process. The present study provides an alternative adsorbent for the highly efficient removal of arsenic from wastewater, which could reduce the human exposure risk to arsenic.

## Figures and Tables

**Figure 1 ijerph-19-11401-f001:**
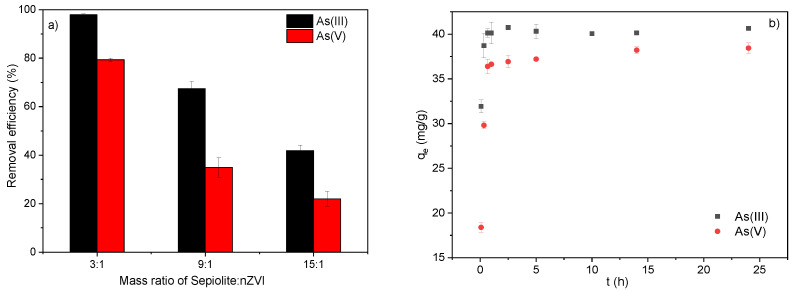
(**a**) Comparison of the As(III)/As(V) adsorption capacity of S-nZVI prepared with different sepiolite:nZVI mass ratios; (**b**) adsorption kinetic curves of As(III) and As(V) by S-nZVI.

**Figure 2 ijerph-19-11401-f002:**
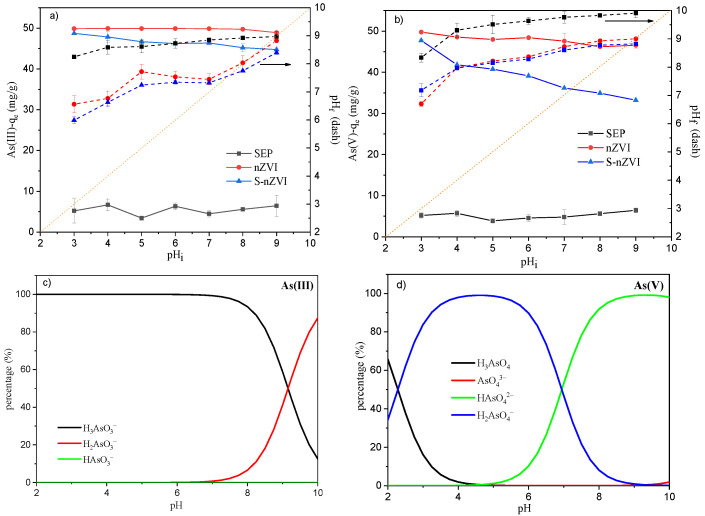
Influence of initial pH on (**a**) As(III) and (**b**) As(V) adsorption capacity (pH_i_ and pH_f_ indicate initial pH before S-nZVI dosing and final pH after adsorption, respectively); the distribution of different (**c**) As(III) and (**d**) As(V) species in aqueous solution calculated with Visual MINTEQ.

**Figure 3 ijerph-19-11401-f003:**
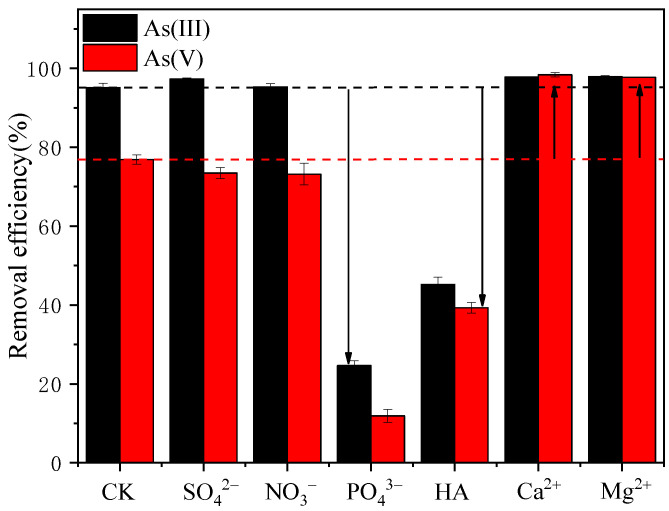
Influence of coexisting ions (SO_4_^2−^, PO_4_^2−^, Ca^2+^, Mg^2+^, and NO_3_^−^) and HA on As(III) and As(V) adsorption (experimental conditions: pH = 7). ‘CK’ stands for ‘control check’, which means no competing ions existed in the aqueous solution.

**Figure 4 ijerph-19-11401-f004:**
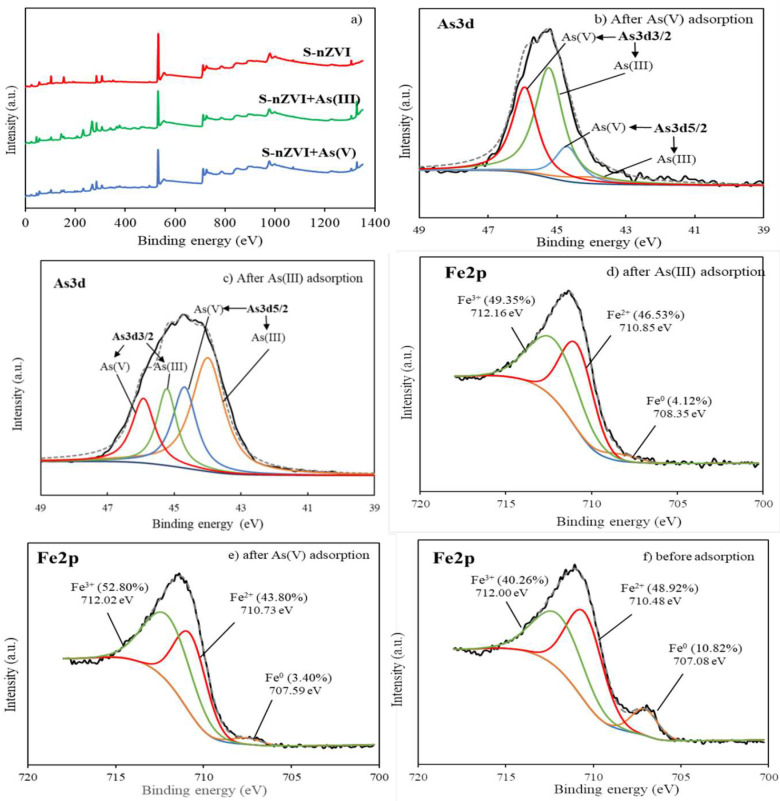
XPS wide-scan spectra (**a**), As3d spectra after As(V) adsorption (**b**) and As(III) adsorption (**c**); and Fe2p spectra after As(III) adsorption (**d**), after As(V) adsorption (**e**), and before adsorption (**f**).

**Table 1 ijerph-19-11401-t001:** Fitted parameters of the pseudo-first-order and pseudo-second-order models for As(III) and As(V) adsorption by S-nZVI.

Kinetic Models	Parameters	As(III)	As(V)
	Q_exp_ (mg/g)	40.71	39.52
Pseudo-first-order kinetics	*k*_1_ (g /mg/ h)	19.00	6.35
	*q_e_* (mg/g)	40.13	37.69
	R^2^	0.998	0.979
Pseudo-second-order kinetics	*k*_2_ (g /mg/ h)	1.119	0.2841
	*q_e_* (mg/g)	40.75	39.01
	R^2^	0.998	0.993

**Table 2 ijerph-19-11401-t002:** Isothermal parameters of As(III) and As(V) adsorption on nZVI and S-nZVI.

	Langmuir Model				Freundlich Model		
	*q_m_* (mg/g)	*K_L_*	*R^2^*	*R_L_*	*K_f_*	1/n	*R^2^*
		As(III)					
nZVI	93.62	0.0037	0.999	0.57–0.98	0.686	0.777	0.995
S-nZVI	165.86	0.090	0.967	0.05–0.69	20.693	0.523	0.867
		As(V)					
nZVI	50.29	0.0085	0.997	0.37–0.96	0.302	0.624	0.974
S-nZVI	95.76	0.044	0.863	0.10–0.82	1.891	0.0346	0.974

## Data Availability

Not applicable.

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
