# Peer review of "Thermodynamics, Kinetics, and Mechanisms of the Co-Removal of Arsenate and Arsenite by Sepiolite-Supported Nanoscale Zero-Valent Iron in Aqueous Solution"

_ijerph, 2022, doi:10.3390/ijerph191811401_

Round 1
Reviewer 1 Report
The study submitted by Ainiwaer et al. deals with the adsorption of As(III) and As(V) by zero-valent iron supported by sepiolite, a porous clay-type mineral found in nature. The paper is very interesting, and the importance of the research is significant due to its potential applications in As contaminated waste water treatment. The study was carried out carefully and the manuscript is well-written and has a very clear and concise structure. Overall, I recommend publication of the manuscript in Environmental Research and Public Health Journal after the consideration of a few revisions (see attachment).

Author Response
Answers to the reviewers:
Reviewer 1
General Review:
Reviewer: The study submitted by Ainiwaer et al. deals with the adsorption of As(III) and As(V) by zero-valent ironsupported by sepiolite, a porous clay-type mineral found in nature. The paper is very interesting, and the importance of the research is significant due to its potential applications in As contaminated waste water treatment. The study was carried out carefully and the manuscript is well-written and has a very clear and concise structure. Overall, I recommend publication of the manuscript in Environmental Research and Public Health Journal after the consideration of a few revisions.
Authors: Thank you for your appreciation of our manuscript. All the questions were answered in the followed details. Please check the revision in our revised manuscript.
Title:
Reviewer: ‘Co-removal’: One thing that did not become clear to me in the entire manuscript is whether As(III) and As(V) are in the same solution (and why they would be, is that naturally relevant that they are present simultaneously?) or investigated separately and then compared to each other. It will be good to clarify that.
- A better title might be ‘Thermodynamics, kinetics, and mechanisms of the Co-Removal of
Arsenate and Arsenite by Sepiolite-Supported Nanoscale Zero-Valent Iron in Aqueous Solution’
(but that is just a suggestion)
Authors: Thank you for you suggestion. We think you gave us a good example and we changed the title as you suggested.
Abstract:
Reviewer: L30: The role of precipitation is not discussed anywhere in the manuscript. I suggest removing this from the abstract
Authors: Thank you for your careful revision. We deleted the precipitation.
Introduction:
Reviewer: L37: Remove ‘The introduction’, ‘the’ main heavy metal contaminant – for me arsenic is ONE main heavy metal contaminant. There are many others that are of great concern.
Authors: Thank you for you careful revision. We deleted the “the introduction”. We also revised the first sentence as you suggested.
Reviewer: L48f.: I suggest briefly discussing the aqueous speciation under the investigated experimental conditions already at this point. Also, consider adding one sentence about the redox stability of As(III) and As(V) under ambient conditions/ investigated pH values?
Authors: Thank you for your suggestion. We add a description in this part as you suggested
Reviewer: L51: which sort of nanoparticles? Arsenic? Adsorbent nanoparticles? Please clarify.
Authors: Thank you for your question. We revised this as “nanoparticle size adsorbent”.
Reviewer: L65: ‘According to’ This sentence is confusing. Consider revision.
Authors: Thank you for your suggestion. This sentence was revised
Reviewer: L74: Please add a sentence or two about sepiolite, so the reader knows what sort of material this is.
Authors: Thank you for your suggestion. We added a description of sepiolite in as you suggested.
Materials and Methods:
Reviewer: L89: 100 µm mesh?, remove ‘All the chemicals were of analytical grade.’ It is redundant with L87
Authors: Thank you for your question. 100 mesh sieve here normally means 0.15 mm sieve. We revised this in the Section 2.1. The redundant sentence was deleted.
Reviewer: - L108: Sample characterization: what about BET measurements? Later on, the specific surface area is given, but it is not mentioned in the experimentals
Authors: Thank you for your question. We added a description of BET test in Section 2.3.
Reviewer: - L132: Did you test for As adsorption on the nylon filters?
Authors: Thank you for your question. The solution were filtrated three the 0.22 aquo-system filter membrane, not nylon filter. We revised the mistake.
Reviewer:- L136: Is Fe:sepiolite the same as S-nZVI? If so, I would avoid two different terms as it is a bit confusing. In L179, it is sepiolite:nZVI. Some clarification may help the reader to understand it more easily.
Authors: Thank you for your question. Fe: sepiolite here means the ratio of Fe and sepiolite in the synthesized material. All the ambiguous expression were revised as you suggested.
Reviewer: - L139: Do you mean ‘aliquotes of the solution’?
Authors: Yes. We revised in the manuscript.
Reviewer: - L179: Does the surface area change with the change in Fe:sepiolite?
Authors: Thank you for your question. We did not exactly measure the surface area of S-nZVI with different sepiolite/Fe ratio. We just tested the effect of S-nZVI with different sepiolite/Fe ratio on arsenic adsorption. We believe this data is enough for us to justify the optimal ratio.
Results:
Reviewer: L189: I don’t understand that. The adsorbed amount of As(V) was lower than for As(III) (As(V) 38 mg/g and As(III) 41 mg/g), so why is As(V) removal more advantageous, should it not be the other way (also cf. Figure 1)? Also, consider adding errorbars here (they are displayed in Figure 1) to show the statistically difference between both values.
Authors: Thank you for pointing out the mistake. We have revised the former sentence and added the value of error bars here accordingly. The new version of the sentence is The equilibrium adsorption amounts were 38.44±0.59 mg/g and 40.66±0.21 mg/g for As(V) and As(III), respectively, consistent with the result (Fig. 1a) that As(III) was more advantageous than As(V) for removal by the S-nZVI adsorbent.
Reviewer: L197: Please give the unit of the adsorption rate here (cf. Table 1)
Authors: We have added the unit of k2 (g/mg/h) to the sentence.
Reviewer: L211: What is there difference between 0.990 and 0.99? The values are not consistent with Table 2.
Authors: Thank you for your question. We have revised the sentence into the right form “it was found that the Freundlich model can better describe the absorption process of S-nZVI for As(V) (R2=0.974) while Langmuir model can better describe the As(III) adsorption by S-nZVI (R2=0.967).”
Reviewer: L215: I suggest giving one or two examples here, so the reader easily gets an idea how much higher it is.
Authors: Thank you for your suggestion. We have added an example as per your suggestion.
Reviewer: L228: Consider moving he sentence about Visual Minteq to the description of the aqueous speciation or removing it.
Authors: Thank you for your suggestion. We have removed the sentence according to your suggestion.
Reviewer: L235: I find ‘sharply’ is a bit stretched here. Consider removing
Authors: Thank you for your suggestion. We have removed ‘sharply’ according to your suggestion.
Reviewer: Figure 2: It is not clear to me what pH f is and which information can be obtained from it. Add explanations of symbols pHi and pHf in caption. Also I recommend adapting the x axes of the plots to the same values so the reader can easily compare between speciation and removal
Authors: Thank you for your suggestion. We have added explanations for pHi and pHf in figure captions and also unify the scale of x axes. Further, we have added explanations for the information which can be obtained from pHi and pHf in Lines 243-248.
Reviewer: L280/Figure3: I recommend given the number of pH value here and also adding the experimental conditions of the experiment in the caption of Figure 3.
Authors: Thank you for your suggestion. We have added the experimental condition in Line 287 and the figure caption of Figure 3.
Reviewer: Also what is CK? Please explain the abbreviation.
Authors: ‘CK’ is the abbreviation for ‘control check’, which means no competing ions existed in the aqueous solution. We have added the explanation in the figure caption.
Reviewer: Why did the authors choose this order of the investigated ions? I suggest grouping ions
enhancing and ions decreasing adsorption, and clearly labelling that in the figure, so it is a bit easier for the reader to grasp the main information quickly
Authors: We have revised figure 3 and grouped the ions according to the enhancing/decreasing of removal efficiency. Also, we have labeled the enhancing/decreasing effect in the figure.
Reviewer: L283: Please add references of ‘some previous studies’.
Answer: We have added 2 references in Line 290 concerning the comparison of different effect on As(III) and As(V) removal by anions (Mg2+, Ca2+).
Reviewer: L288: Where was it demonstrated? In this study or in literature? Please add reference.
Authors: We have provided SEM images of S-nZVI in the Supporting Information. From the SEM images we can see that spherical nZVI was well dispersed on the sepiolite surface.
Reviewer: L289: Add determination of specific surface areas to Experimentals and give values earlier in the manuscript.
Authors: The determination method and values of specific surface areas have been added in the Experimental Section (Lines 117-120).
Reviewer: L290: I don’t understand which results confirm that no agglomeration took place. In this context, it would be useful to show the SEM images in the Supporting Information.
Authors: Thank you for pointing out the weakness. We have provided SEM images of S-nZVI in the Supporting Information. From the SEM images it can be seen that spherical nZVI was well dispersed on the sepiolite surface without agglomeration.
Reviewer: L298: The authors state that the relative abundance of each valence state can be calculated from the comparison of the areas of the subpeak. To me, the area of the green peak (As(III)) is similar to the red peak (As(V)), but the authors state that only 19 % is detected as As(III). Please explain.
Authors: Here we will demonstrate the calculation process of the As(III) or As(V) ratio based on the peak area comparison. The following chart shows the area for each sub-peaks of the As3d spectra. Therefore, the ratio of As(III) was calculated by the following equation:
As(III)%=(SBE=44.00+SBE=45.23)/(SBE=44.00+ SBE=45.23+ SBE=45.39+ SBE=44.70)×100%
As(V)%=( SBE=45.39+ SBE=44.70)/(SBE=44.00+ SBE=45.23+ SBE=45.39+ SBE=44.70)×100%
binding energy (eV) |
Peak attribution |
As valence state |
After As(III) adsorption (Fig. 4b) |
After As(III) adsorption (Fig. 4c) |
||
area |
ratio% |
area |
ratio% |
|||
44.00 |
As3d5/2 (orange) |
As(III) |
20574.66 |
60.36 |
1336.63 |
53.55 |
45.23 |
As3d3/2 (green) |
8522.45 |
10536.90 |
|||
44.70 |
As3d5/2 (blue) |
As(V) |
10881.13 |
39.64 |
3014.73 |
46.45 |
45.93 |
As3d3/2 (red) |
8226.58 |
7284.91 |
And here we apologize for the careless mistake when adding the areas of the 2 sub-peaks. We have revised the ratios of As(III) and As(V) throughout the section and replaced them with the correct calculation results. We are sure that the results are in correct form in the present version.
Reviewer: L304: please add reference of ‘the previous study’
Authors: We have added the relevant reference in Line 336.
Reviewer: L320: Are errorbars for the ratios available?
Authors: Sorry but the error bars for ratios are not available because the ratios were calculated based on the peak area comparisons.
Conclusion:
Reviewer: L356: I think Cd(II) is not mentioned anywhere else in the manuscript, consider removing
Authors: Thank you for pointing out the error. We have removed the mistaken word here.
Reviewer: L363: This can probably generalized towards all cations?
Authors: Thank you for your question. In our study, only Ca2+ and Mg2+ were chosen as the co-existing cations to test their effect on arsenic adsorption. Although the results of our study demonstrated that Ca2+/Mg2+ enhanced As(V) adsorption but little affected As(III) adsorption, this conclusion may not be suitable for other cations. For example, it was reported in a previous study that Fe2+ can suppress arsenite adsorption because Fe2+ tends to form complexes with arsenite, leading to arsenite species with less deprotonated form [Yasinta et al., 2018; doi.org/10.1155/2018/3975948]. In conclusion, if the cation is more affinitive to the adsorbent surface and can act as a bridge between adsorbent and arsenic, the arsenic adsorption amount will be enhanced. Otherwise, if the cation is more affinitive to arsenic species, it will suppress the deprotonation of arsenic and decrease the adsorption amount.
Reviewer: Typos and Formatting Issues:
- L28,29, 358 : Change As( V ) to As(V)
- Upper and lower characters:
o L100: NaBH4, L129f, L145:lower case of symbols in equations, L172: NO3- o L209: R2
o L290: m2
o L311: Fig 4d-f
Authors: Thank you for pointing out the mistakes. We have revised the typos and formatting errors to guarantee the correctness of our manuscript.

Reviewer 2 Report
From the reviewer's point of view, in the introduction it is desirable to unite the references by their semantic subject matter, rather than to consider the work each separately - then the introduction will be perceived more clearly.
Can the obtained results be interpreted using Dubinin-Radushkevich and Dubininin-Astakhov equations, or only the Langmuir and Freundlich models sre applicable here?
Author Response
Reviewer 2
Reviewer: From the reviewer's point of view, in the introduction it is desirable to unite the references by their semantic subject matter, rather than to consider the work each separately - then the introduction will be perceived more clearly.
Authors: Thank you for your suggestion. We revised as you suggested to make the introduction more clearly.
Reviewer: Can the obtained results be interpreted using and Dubininin-Astakhov equations, or only the Langmuir and Freundlich models are applicable here?
Authors: Thank you for your good suggestion. We have added contents related to the fitting of As(III)/As(V) adsorption isotherms with D-A and D-R equations. We found that the results can be well fitted with both models (R2>0.95), which means that adsorption of As(III)/As(V) on S-nZVI happened in homogeneous microporous systems [Dubinin and Radushkevich, 1947; Chen and Yang,1994]. The relevant description of D-A and D-R modeling can be found in Section 2.4.3, and the discussion in regard to the model fitting was added in Section 3.2. The fitting results can be found in Figure S2 and Table S2.

Round 2
Reviewer 1 Report
Thank you very much for addressing my questions and concerns.